# Systematic review and meta-analysis on the prevalence and associated factors of depression among hypertensive patients in Ethiopia

Worku Chekol Tassew[1]*, Getanew Kegnie Nigate[2], Getaw Wubie Assefa[3], Agerie Mengistie Zeleke[4], Yeshiwas Ayal Ferede[2]

1 Department of Medical Nursing, Teda Health Science College, Gondar, Ethiopia, 2 Department of Reproductive Health, Teda Health Science College, Gondar, Ethiopia, 3 Department of CTID &HIV Medicine, Teda Health Science College, Gondar, Ethiopia, 4 Department of Clinical Midwifery, Teda Health Science College, Gondar, Ethiopia

* workukid16@gmail.com

## Abstract

### Background

Identifying individuals at increased risk for depression allows for earlier intervention and treatment, ultimately leading to better outcomes and potentially preventing severe symptoms. However, to date, no systematic reviews or meta-analyses have estimated the prevalence of depression among hypertensive patients. Thus, this review was initiated to determine the prevalence and factors associated with depression among patients with hypertension in Ethiopia.

### Methods

Multiple databases, such as PubMed, African Journals Online, the Cochrane Library, and Google Scholar, were used to ensure wider coverage of relevant studies. The data extracted from Microsoft Excel were imported into STATA version 11 (Stata Corp LLC, TX, USA) for further analysis. The pooled prevalence of depression was estimated using a random effects model. To evaluate statistical heterogeneity, the Cochrane Q test and $I^2$ statistic were used.

### Results

The random effect model indicated that the pooled prevalence of depression in 12 studies conducted in Ethiopia was 32.43% (95% CI: 25.18, 39.67%). Being female (POR = 2.41; 95% CI: 1.89, 3.07, $I^2$ = 17.7%, P = 0.302), having comorbid illnesses (POR = 3.80; 95% CI: 2.09, 6.90, $I^2$ = 81%, P = 0.005), having poor blood pressure control (POR = 3.58; 95% CI: 2.51, 5.12, $I^2$ = 0.0%, P = 0.716), having a family history of depression (POR = 3.43; 95% CI: 1.98, 5.96, $I^2$ = 62.6%, P = 0.069), being single (POR = 2.30; 95% CI: 1.35, 3.99, $I^2$ =

**Data Availability Statement:** All relevant data are within the manuscript and its Supporting Information files.

**Funding:** The author(s) received no specific funding for this work.

**Competing interests:** The authors have declared that no competing interests exist.

**Abbreviations:** CI, Confidence interval; HTN, Hypertension; NCD, Non-Communicable Disease; NOS, Newcastle–Ottawa scale; POR, Pooled **odds ratio**; PRISMA, Preferred Reporting Items for Systematic Reviews and Meta-Analysis.

48.0%, P = 0.146) and having poor social support (POR = 4.24; 95% CI: 1.29, 13.98, $I^2$ = 95.8%, P<0.001) were positively associated with depression among hypertensive patients.

## Conclusion

Overall, the results of our review showed that depression affects a significant number of Ethiopians who have hypertension. Being female, being single, having comorbidities, having poor blood pressure control, having a family history of depression, and having poor social support were factors associated with depression among patients with hypertension. For those who are depressed, improving the psycho-behavioral treatment linkage with the psychiatric unit can result in improved clinical outcomes.

## Trial registration

**Prospero Registration number**: CRD42024498447. https://www.crd.york.ac.uk/prospero/display_record.php?ID=CRD42024498447.

## Background

According to the latest data from the World Health Organization (WHO), non-communicable diseases (NCDs) are indeed a major global health challenge, claiming the lives of 41 million people each year. This translates to 86% of all deaths globally, making NCDs the leading cause of mortality worldwide [1]. By 2025, NCDs are projected to account for over 70% of all deaths globally, highlighting their significant impact on public health [2]. The fact that 41 million people could die from NCDs by 2025, primarily due to preventable conditions such as CVD (48%), cancer (21%), and diabetes (3%), highlights the urgent need for effective prevention strategies worldwide [3]. With an estimated 1.13 billion individuals worldwide having hypertension and staggering two-thirds residing in LMICs, this condition poses a major public health challenge [4]. The increase in hypertension (HTN) in LMICs compared to its stability or decrease in developed nations is a concerning and well-documented trend in global health research [5].

Representing 7.4% of the global burden of disease, mental and behavioral disorders affect a vast number of people worldwide [6]. With 350 million people currently experiencing depression and a lifetime risk of 7%, it is clear that depression affects a significant portion of the global population. The World Health Organization's prediction that depression will become the leading cause of disability globally by 2030 highlights the severity of its impact. Depression can significantly impair a person's ability to work, maintain relationships, and function in daily life [7]. Multiple studies have reported greater rates of depression in individuals with HTN than in those with normal blood pressure. HTN can cause chronic stress in the body, impacting the hypothalamic–pituitary–adrenal (HPA) axis, which regulates stress hormones and can contribute to mood disturbance [8]. The combination of hypertension (HTN) and depression poses significant challenges on multiple levels, making it crucial to address both conditions simultaneously for optimal health and well-being [9].

The presence of depression alongside hypertension, known as depression comorbidity, can have significant negative consequences for patients [10], more expensive medical care [11], a lower rate of treatment adherence [12], and even a higher death rate [13]. Depression can have a significant impact on various aspects of a person's life, including social interactions and

work performance [14,15]. The finding that patients with hypertension have a greater risk of depression than does the general population is well established and well documented [16]. Low educational attainment, body mass index, poverty, absence of social support, and patient residency are some of the factors associated with depression among hypertensive patients [17–19].

Comorbid mental illnesses such as depression pose a significant challenge and are often misdiagnosed and ineffectively managed [20]. Investigating the psychological aspects of hypertension and related disorders in Ethiopia is critical due to the increasing prevalence and mortality rates, coupled with inconsistencies in existing international research. Identifying individuals at increased risk for depression allows for earlier intervention and treatment, ultimately leading to better outcomes and potentially preventing severe symptoms. By identifying the risk factors associated with depression, policies and programs can be designed to target those most vulnerable individuals, maximizing their effectiveness and efficiency. However, to date, no systematic reviews or meta-analyses have estimated the prevalence of depression among hypertensive patients. Thus, this review was initiated to determine the prevalence and factors associated with depression among patients with hypertension in Ethiopia.

## Methods

### Search strategies

The review adheres to the Preferred Reporting Items for Systematic Reviews and Meta-Analysis (PRISMA) standards S1 File [21]. Multiple databases, such as PubMed, African Journals Online, the Cochrane Library, and Google Scholar, were used to ensure wider coverage of relevant studies. Focusing on research specifically reporting prevalence of depression among hypertension patients in Ethiopia allows for a specific and targeted analysis. MeSH (Medical Subject Heading) phrases were combined to create a search strategy for each database. The following search terms were used: "depression" or "depressive disorder" AND "hypertension" or "elevated blood pressure" or "high blood pressure" AND "associated factors" or "determinants" or "risk factors "AND "Ethiopia".

### Inclusion criteria and exclusion criteria

We included all studies published in the English language in Ethiopia that reported the prevalence of depression and/or its associated variables in patients with hypertension and that were published as of December 30, 2020. Studies that focused on particular demographic characteristics, case reports, case series, letters to the editor, and studies that did not report the prevalence of depression were excluded.

**Outcome of review.** The outcome variable of the review is the prevalence of depression and the factors associated with it among individuals with hypertension.

### Study selection

All of the studies that were identified using different electronic databases were exported into Endnote X7. After duplicate articles were eliminated, the abstract and full text of each article were separately examined by two authors (WCT and YAF), who then screened all of the articles for eligibility.

### Quality assessment

The Newcastle–Ottawa scale (NOS), which was modified for cross-sectional studies, was used to evaluate the quality of each publication [22]. Three major components make up the tool.

The tool's first section, which rates each study's methodological quality with five stars, evaluates the following areas: sampling procedure, sample size, response rate, and identification of the risk factor or exposure. The study's comparability is rated with a potential of two stars in the second section of the tool. The final part of the tool assigns a potential three stars based on the results and statistical analysis of the original investigation. Two reviewers (WC and YAF) independently assessed the quality of the included studies; any differences in the two reviewers' assessments of quality were synchronized by the third author (GWA).

### Data extraction and management

The prevalence and associated variables of depression among hypertension patients from each study were summarized by three authors (WCT, GKN, & YAF) using the data extraction format developed with the help of the Joanna Briggs Institute (JBI) data extraction tool for prevalence studies. For every study, the following information was retrieved: the name of the first author, the year the study was published, the sample size, the study's design, the prevalence of depression and associated factors, and the confidence intervals.

### Outcome measurement

The dependent variable of the review was depression in hypertension patients; in the original research, this dependent variable was assessed using the Ethiopian-validated Patient Health Questionnaire Nine (PHQ-9) [23].

### Statistical analysis

The data extracted from Microsoft Excel were imported into STATA version 11 (Stata Corp LLC, TX, USA) for further analysis. The pooled prevalence of depression was estimated using the random effects model [24]. To evaluate statistical heterogeneity, the Cochrane Q test and $I^2$ statistic were used [25]. To reduce the variance of point estimates between primary studies, subgroup analysis by region was carried out. Furthermore, sensitivity analysis was performed to determine the impact of specific studies on the pooled estimate.

To investigate publication bias (small study effect), a funnel plot and Egger's test were utilized statistically [26]. To determine the factors associated with depression in hypertension patients, the pooled odds ratio (POR) and 95% confidence interval were computed.

**Systematic Review registration:** The review has been recorded under PROSPERO with the number CRD42024498447.

## Results

### Description of the included articles

**A total of** 989 articles were found in the initial search and imported into the citation manager program EndNote X 7.0. Research was retrieved from African Journals Online (186), Google Search (151), African Journals Online (526), and the Cochrane **Library** (126). **Of these**, 316 studies that were duplicates were found and removed. Twelve studies were included **in** the final analysis after the articles were reviewed and the eligibility criteria were met (Fig 1).

Of the 12 articles, three were from Addis Ababa [27–29], three from southern Ethiopia [30–32], three from the Amhara region [33–35], two from the Oromia region [36,37] and one from Harari [38]. Regarding study designs, all studies were institution-based cross-sectional studies (Table 1).

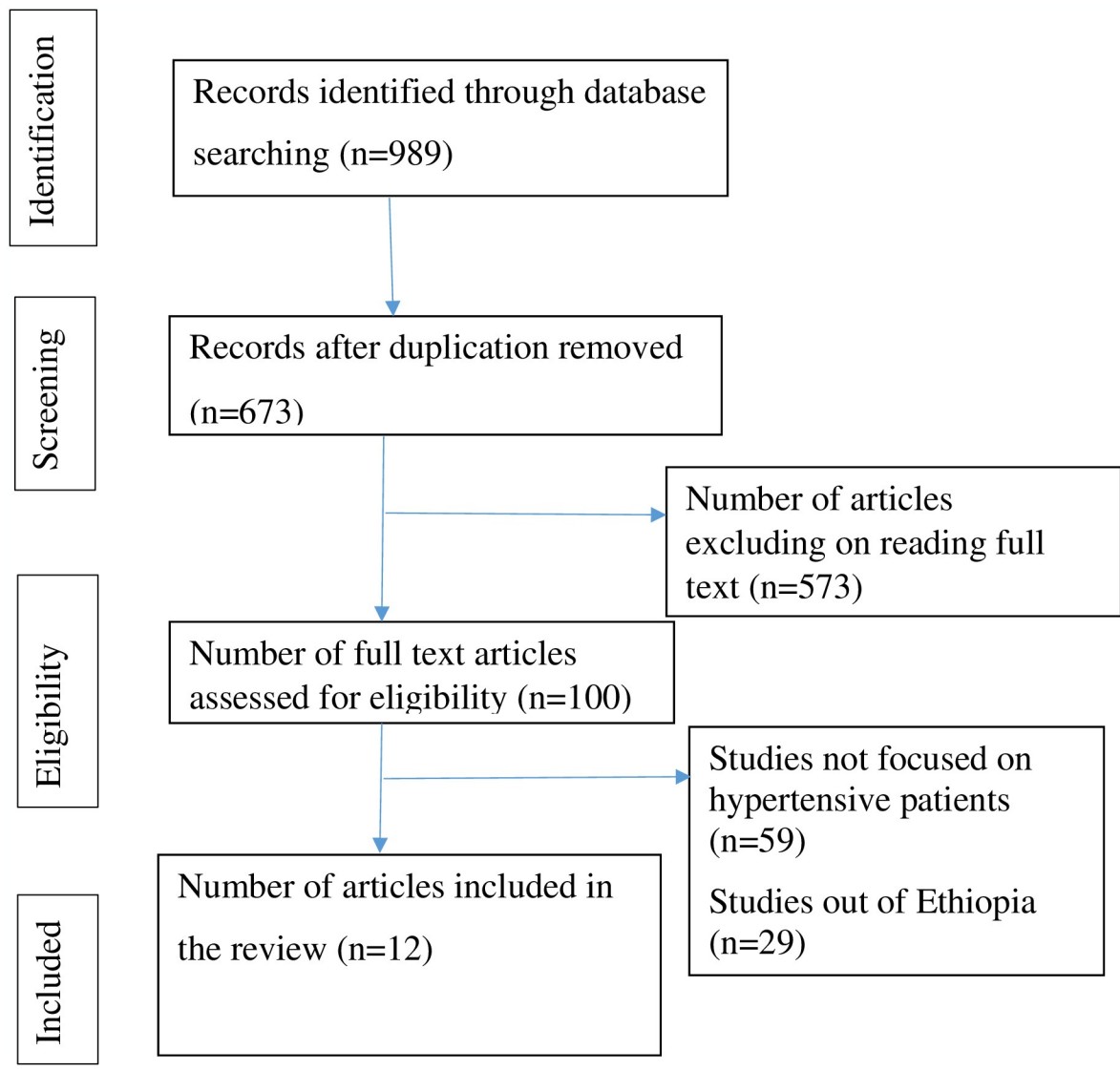

**Fig 1. PRISMA flow diagram for a systematic review and meta-analysis of depression and associated factors among hypertensive patients in Ethiopia.**

### Pooled estimate of depression among people living with hypertension in Ethiopia

The random effect model indicated that there was moderate heterogeneity among the studies (I2 = 59.2%, p = 0.005), and the pooled prevalence of depression in 12 studies conducted in Ethiopia was found to be 32.43% (95% CI: 25.18, 39.67%)(Fig 2).

### Handling heterogeneity

We estimated the pooled prevalence using the random-effects model because the included studies had a moderate level of heterogeneity. Meta-regression analysis, sensitivity analysis, and subgroup analysis were used to address this issue. In the sensitivity analysis, no influential studies were found (Fig 3).

**Table 1. Characteristics of the included studies and prevalence of depression.**

| Author | Pub Year | Region | Study Design | Sample size | Prevalence | Quality score |
|---|---|---|---|---|---|---|
| Asmare et al. [27] | 2022 | Addis Ababa | Cross sectional | 416 | 37.8 | High |
| Esubalew T & Samual A [33] | 2021 | Amhara | Cross sectional | 239 | 19.49 | High |
| Afework E & Caridad O [34] | 2020 | Amhara | Cross sectional | 404 | 5.73 | Medium |
| Abdisa et al. [38] | 2022 | Harar | Cross sectional | 491 | 27.2 | High |
| Nigusu et al. [30] | 2023 | South | Cross sectional | 315 | 37.1 | Medium |
| Gebre et al [31] | 2020 | South | Cross sectional | 295 | 24.7 | High |
| Umer et al [36] | 2019 | Oromia | Cross sectional | 293 | 46 | High |
| Alemayehu et al [28] | 2022 | Addis Ababa | Cross sectional | 424 | 43.6 | High |
| Yazew et al [35] | 2019 | Amhara | Cross sectional | 422 | 51.1 | High |
| Soboka et al [37] | 2017 | Oromia | Cross sectional | 401 | 31.60 | High |
| Ayalew et al. [32] | 2023 | South | Cross sectional | 633 | 36.01 | High |
| Assefa et al. [29] | 2016 | Addis Ababa | Cross sectional | 479 | 17.57 | High |

We also conducted subgroup analysis and reported estimates, taking into account additional potential sources of variation, such as region (Fig 4).

Subgroup analysis revealed a prevalence of 39.30% (95% CI: 25.22, 53.38) in the Oromia region, while the Amhara region reported the lowest prevalence (26.05% 95% CI: -1.75, 53.84).

## Meta-regression

We further fitted a meta-regression using the random effects model on the aggregated study-level variables to address heterogeneity. According to the univariable meta-regression analysis, the number of participants, sample size, and publication year were not significantly associated with depression (Table 2).

**Quality assessment of the included studies.** Using the Newcastle–Ottawa scale, a tool modified for cross-sectional studies, the quality of each original study was evaluated critically. Approximately ten (n = 883.3%) of the included studies were of high quality, and the remaining two (n = 16.6%) were of medium quality, according to the quality assessment summary of all the included studies S2 File.

## Publication bias

Publication bias is evident from the asymmetric distribution displayed in the funnel plot and the significant Egger's test result (P< 0.001) (Fig 5).

## Factors associated with depression among hypertensive patients in Ethiopia

The pooled result of four studies in the random effect model of the meta-analysis of identified associated factors revealed that among patients with hypertension, being female was strongly associated with depression (POR = 2.41; 95% CI: 1.89, 3.07; $I^2$ = 17.7%; P = 0.302). (Fig 6).

The pooled effect of three studies in the random effect model of the meta-analysis of identified associated factors revealed that people with comorbid illnesses (POR = 3.80; 95% CI: 2.09, 6.90, $I^2$ = 81%, P = 0.005) had a 3.80-fold increased risk of depression compared to people without any comorbid illnesses. Because the included articles were heterogeneous, a random effect model was employed (Fig 7).

According to the pooled odds ratio of the two investigations, people with poor blood pressure control had a 3.58-fold increased risk of depression compared to people with good blood

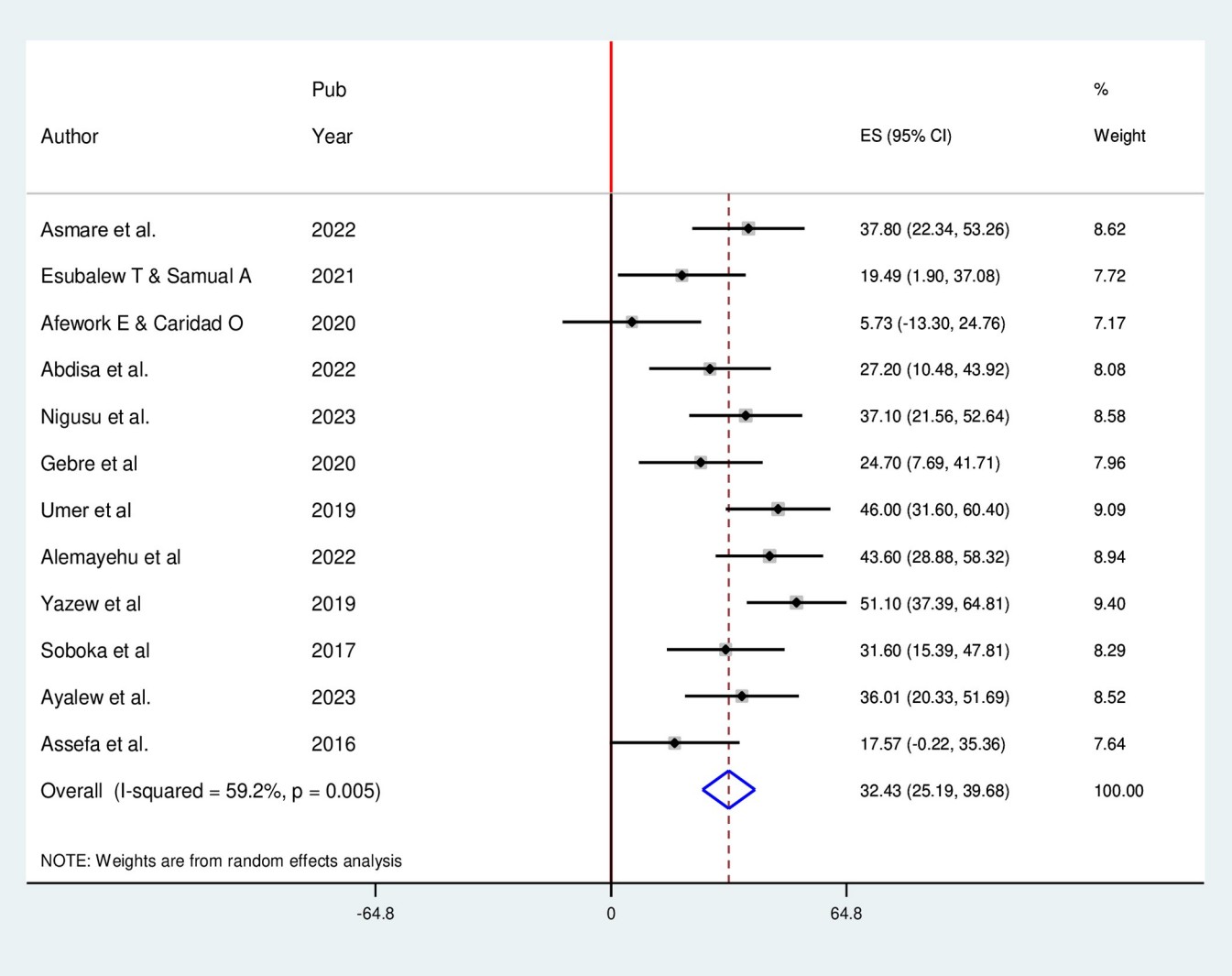

**Fig 2. Forest plot for the meta-analysis of depression among hypertensive patients (n = 12).**

pressure control (POR = 3.58; 95% CI: 2.51, 5.12; $I^2$ = 0.0%; P = 0.716). Because there was no heterogeneity among the included articles, a fixed effect model was employed. A significant association was found between the occurrence of depression and a family history of the condition in three different investigations. Individuals who had a positive family history of depression were 3.43 times more likely to be depressed than those who did not (POR = 3.43; 95% CI: 1.98, 5.96; $I^2$ = 62.6%, P = 0.069). The finding of variability among the included publications led to the use of the random effect model (Fig 8).

Three studies' random pooled effects showed that there was a statistically significant correlation between depression and marital status. The results showed that patients (POR = 2.30; 95% CI: 1.35, 3.99; $I^2$ = 48.0%, P = 0.146) who experienced depression alone were 2.30 times more likely to have depression than patients who were married. A significant association was found between depression and low social support, as demonstrated by the random pooled effect of five studies. Compared with patients with strong or good social support, patients with

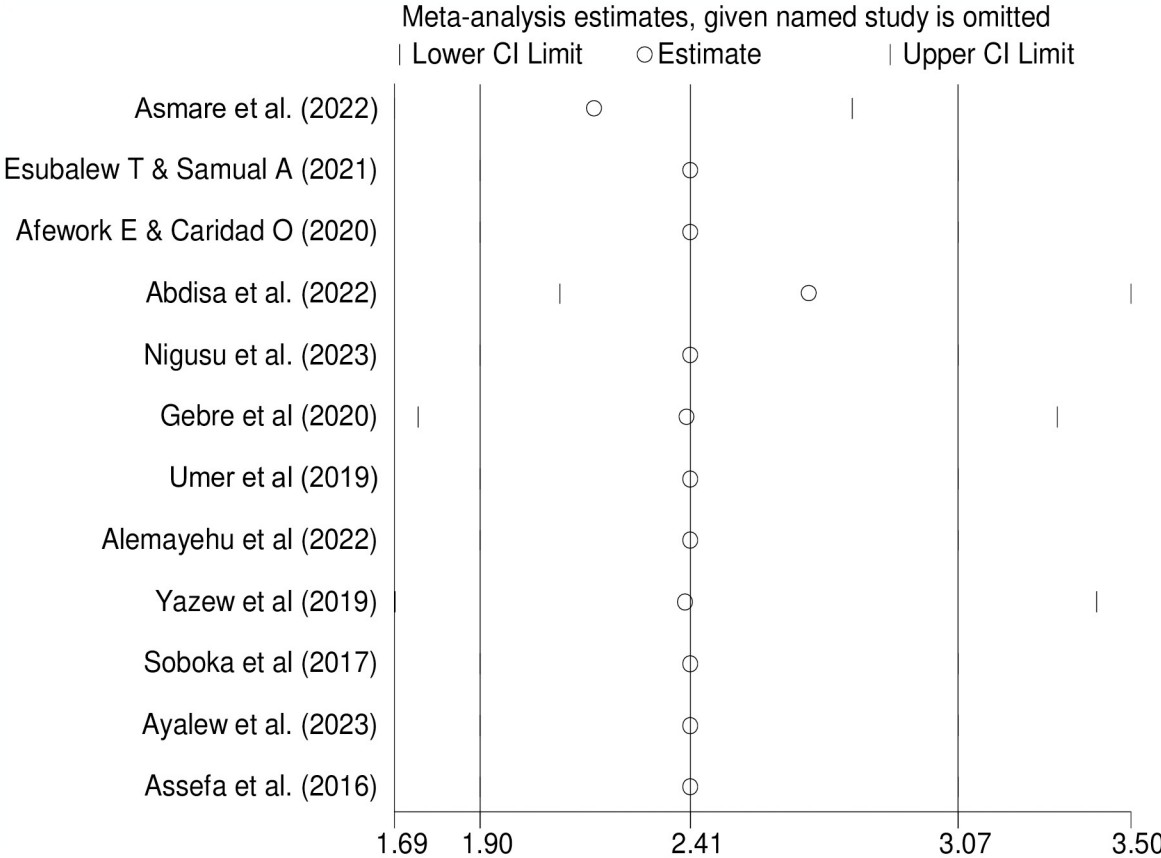

**Fig 3. Sensitivity analysis of studies included in the meta-analysis.**

poor social support were 4.24 times more likely to be depressed (POR = 4.24; 95% CI: 1.29, 13.98; $I^2$ = 95.8%, P<0.001) (Fig 9).

## Discussion

Coexistence of both depression and hypertension could be explained by depression as a consequence of hypertension or as a risk factor for developing hypertension, or the two conditions may have common pathophysiology and manifest together; however, their temporal and causal relationship remains still unclear. Both diseases are commonly found to coexist, and treatment for hypertension has been reported to affect depression and vice versa. Different factors should be taken into account to examine an independent association between BP and depression.

In Ethiopia, pooled estimates of the magnitude of depression among hypertensive patients are lacking. This is the first systematic review on the magnitude of depression in people with hypertension. According to the current meta-analysis, people with hypertension frequently experience depression. Across 12 studies conducted in Ethiopia, the combined prevalence of depression in hypertension patients was 32.43% (95% CI: 25.18, 39.67%). This result is in line with a Chinese study that reported a summary prevalence of depression of 26.8% (95% CI: 21.7%–32.45%) among hypertension patients [39].

However, the results of this review are less than those of a systematic review carried out in Saudi Arabia, which indicated that 57.3% of hypertension patients had an overall prevalence of

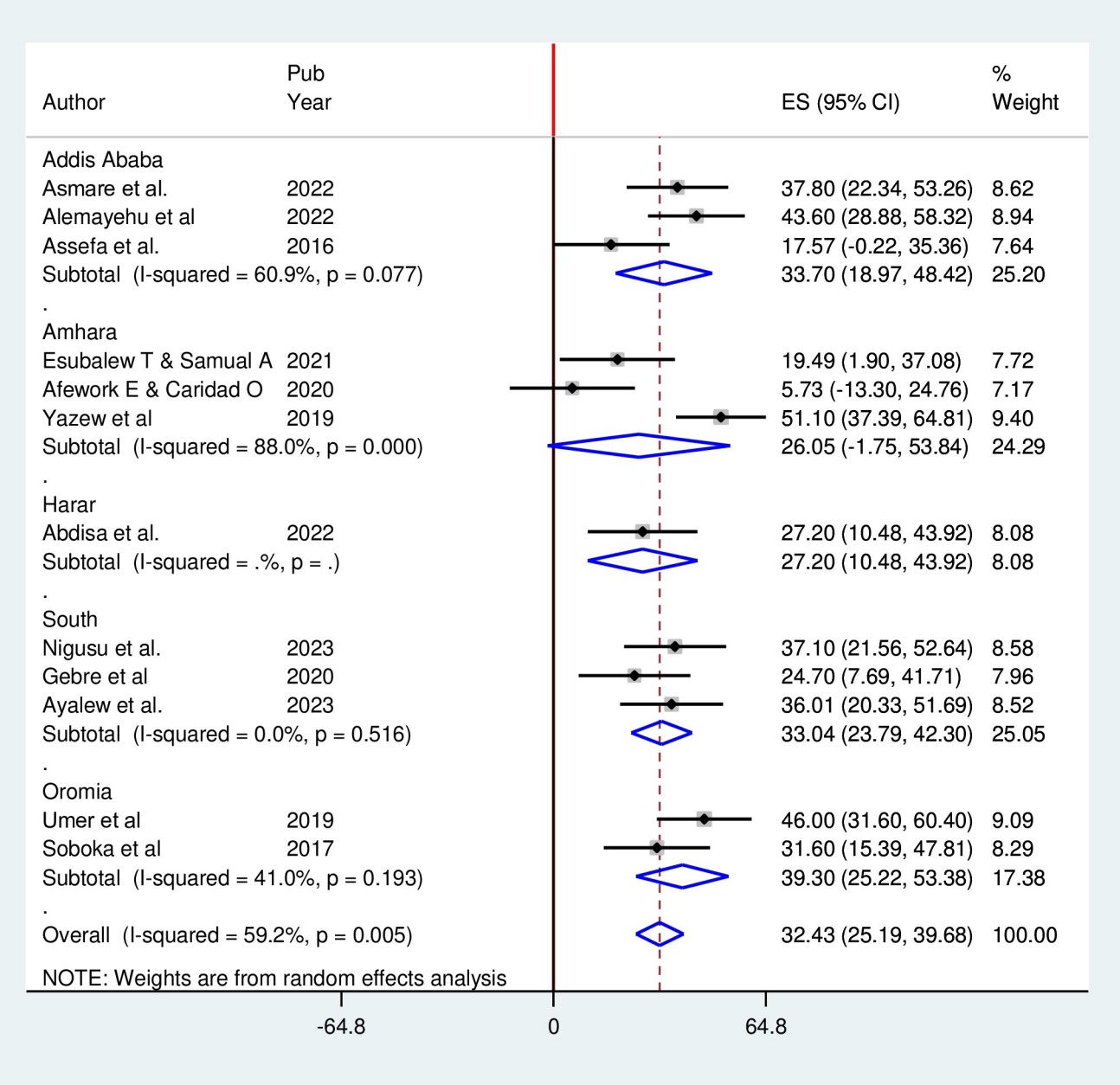

**Fig 4. Subgroup analysis (by region) of studies included in the meta-analysis on depression among hypertensive patients in Ethiopia.**

**Table 2. Univariable meta-regression analysis results for the prevalence of depression among hypertensive patients in Ethiopia.**

| Study level variables | Coefficients | Standard error | P>|t| | [95% CI] |
|---|---|---|---|---|
| Participants | 0.11 | 0.15 | 0.48 | (-0.24–0.47) |
| Pub year | -0.19 | 0.91 | 0.83 | (-2.30–1.91) |
| Sample size | -0.11 | 0.14 | 0.46 | (-0.44–0.22) |

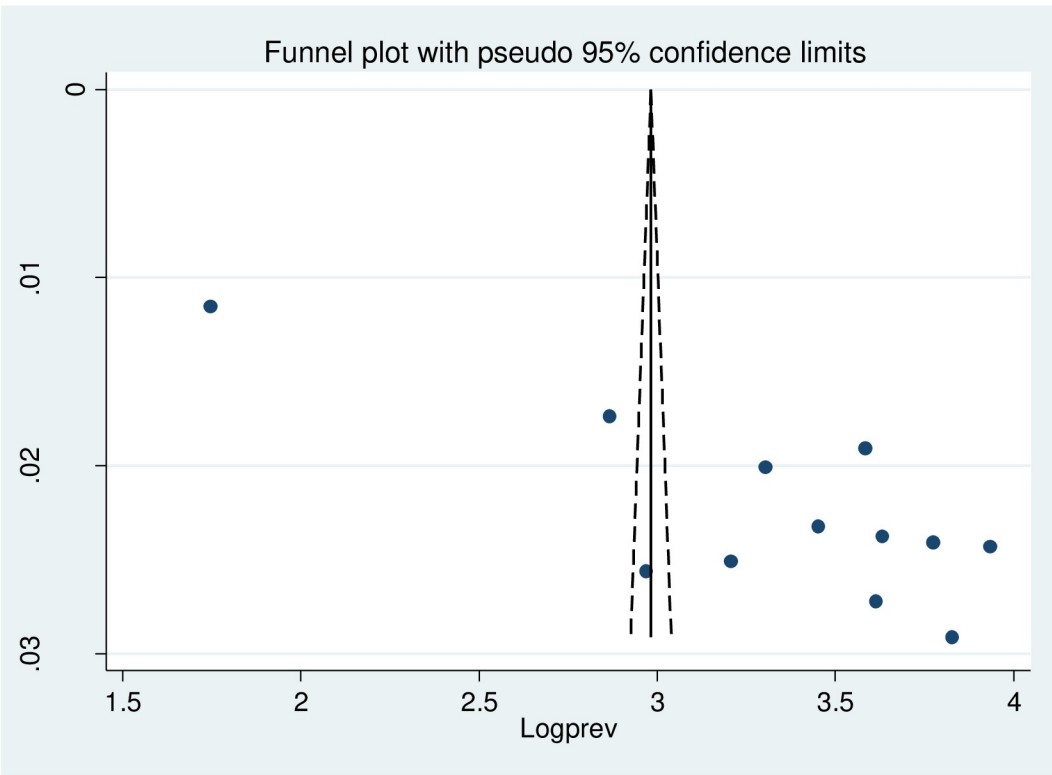

**Fig 5. Funnel plot testing publication bias (random, N = 12).**

depression [40]. However, the results of this review are greater than those of a comprehensive review conducted in Spain, which revealed that the prevalence of depression in hypertension patients was 18% (95% CI = 13–24), [41], Afghanistan 58.1% and 65.8% [42,43] and in Northwest China, which reported that the prevalence of depression in hypertensive patients was 14.5% [9]. There could be a number of reasons for this disparity, including differences in healthcare access, cultural factors, and environmental stressors across diverse locations within Ethiopia, which could contribute to varying depression prevalence.

The pooled results of four studies in the random effect model of the meta-analysis of identified associated factors revealed that among patients with hypertension, being female was strongly associated with depression. This finding is consistent with those of studies performed in Ethiopia involving the general population [44,45]. The explanation for this could be that changes in estrogen and progesterone levels during a woman's life cycle, such as those that occur during menopause, pregnancy, and puberty, have direct effects on mood and increase the risk of depression [46].

The pooled effect of three studies in the random effect model of the meta-analysis of identified associated factors revealed that people with comorbid illnesses had an increased risk of depression compared to people without any comorbid illnesses. These findings are similar to those of studies performed in the USA and Ethiopia [45,47]. Comorbidities and depression have a complicated and multidimensional relationship. Although the direct effects of inflammatory and metabolic alterations on the neurological system are well established, more research is needed to determine the independent contributions of particular parameters, such as systolic blood pressure and fasting blood glucose, to depression [48,49].

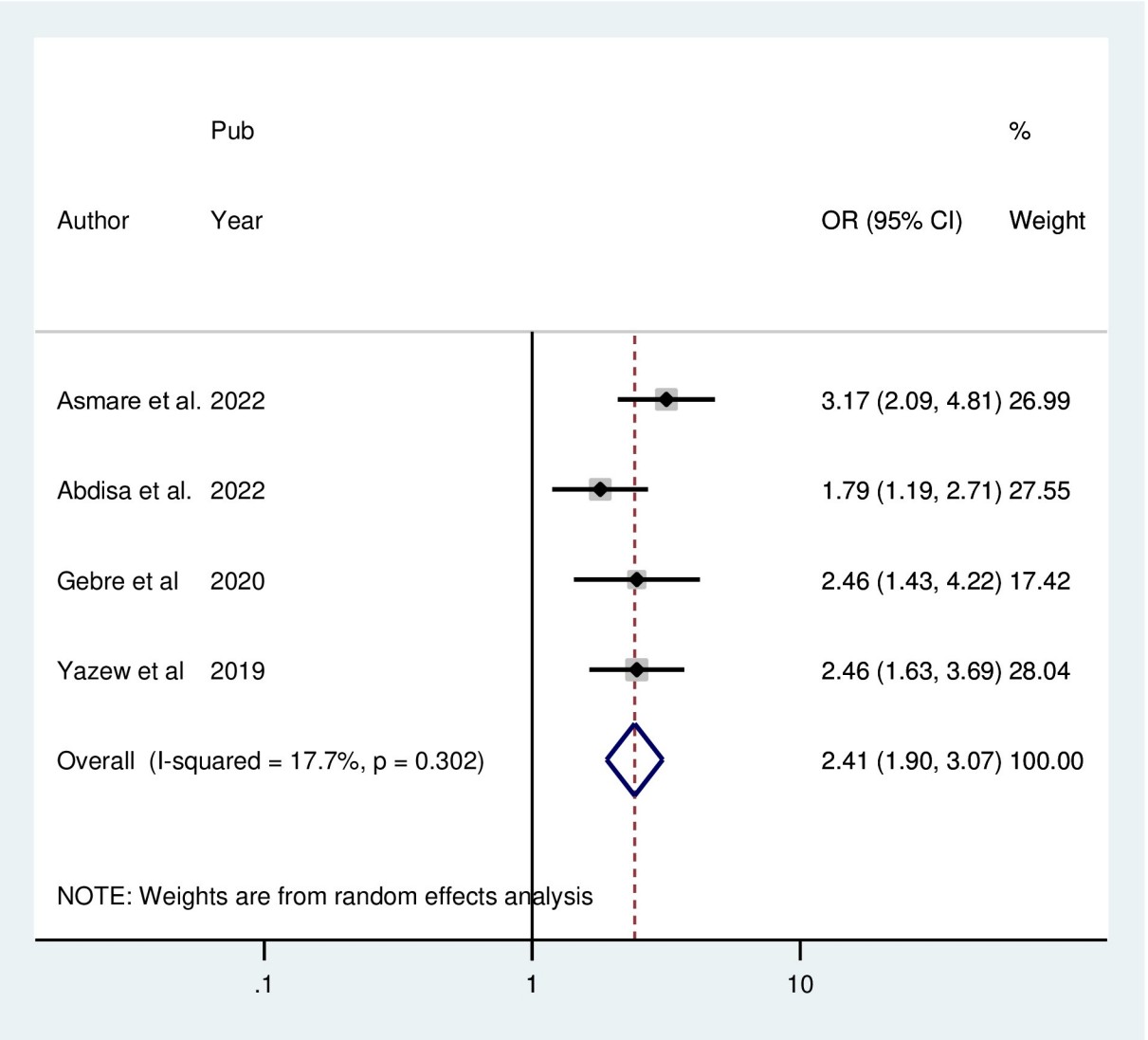

**Fig 6. Forest plot for the association between female sex and depression among hypertensive patients in Ethiopia.**

According to the pooled odds ratio of the two investigations, people with poor blood pressure control had an increased risk of depression compared to people with good blood pressure control. This might be because people who do not take their medications as directed have poor blood pressure control, which in turn leads to depression [50]. Individuals who have a positive family history of depression are more likely to be depressed than those who do not. This finding is supported by a study performed in Ethiopia [51]. The reason may originate from non-biological or biological perspectives, such as shared familial stressors and environmental factors [52]. According to our findings, patients who were single were more likely to report depression than were married patients. This could be because variables that underlie both being single and depression, such as social isolation, loneliness, or financial difficulties, produce an erroneous association [50].

According to the findings, patients who had poor social support were more likely to be depressed than patients who had strong or good social support. This finding is supported by

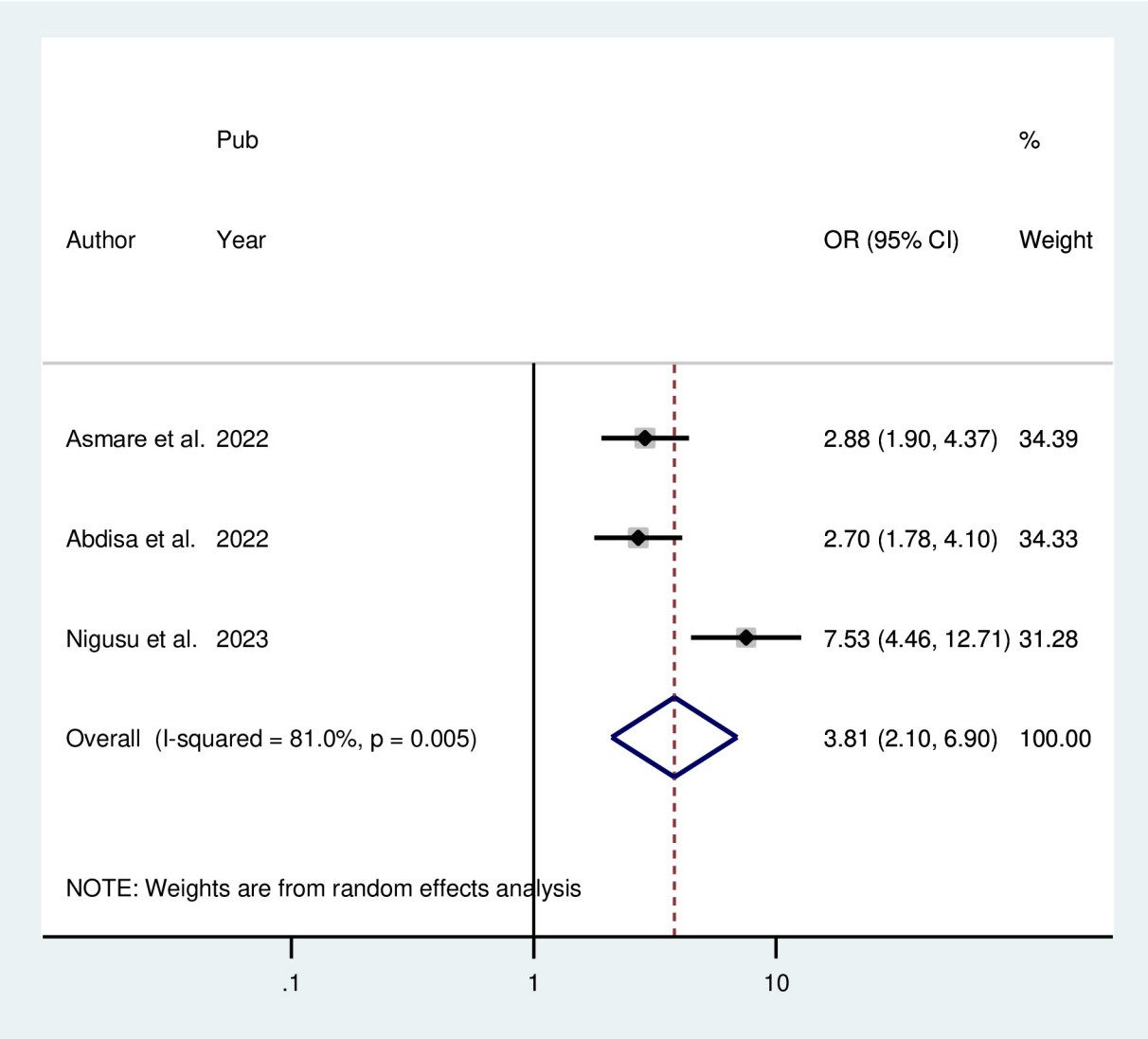

**Fig 7. Forest plot for the association between comorbid illnesses and depression among hypertensive patients in Ethiopia.**

studies performed in Ethiopia [44,45]. It is true that stress levels can rise sharply in response to perceived loneliness and lack of support, which increases the likelihood of developing depression. Strong social networks, on the other hand, can provide a sense of community, practical help, and emotional understanding, acting as a buffer against stress and depression [53].

## Strengths and limitations

Although sensitivity, subgroup, and meta-regression analyses were used to lessen the influence of heterogeneity, the degree of heterogeneity among the included studies was significant. Despite these drawbacks, this first systematic review and meta-analysis on the magnitude of depression in people living with hypertension in Ethiopia provided a clear summary of the existing knowledge.

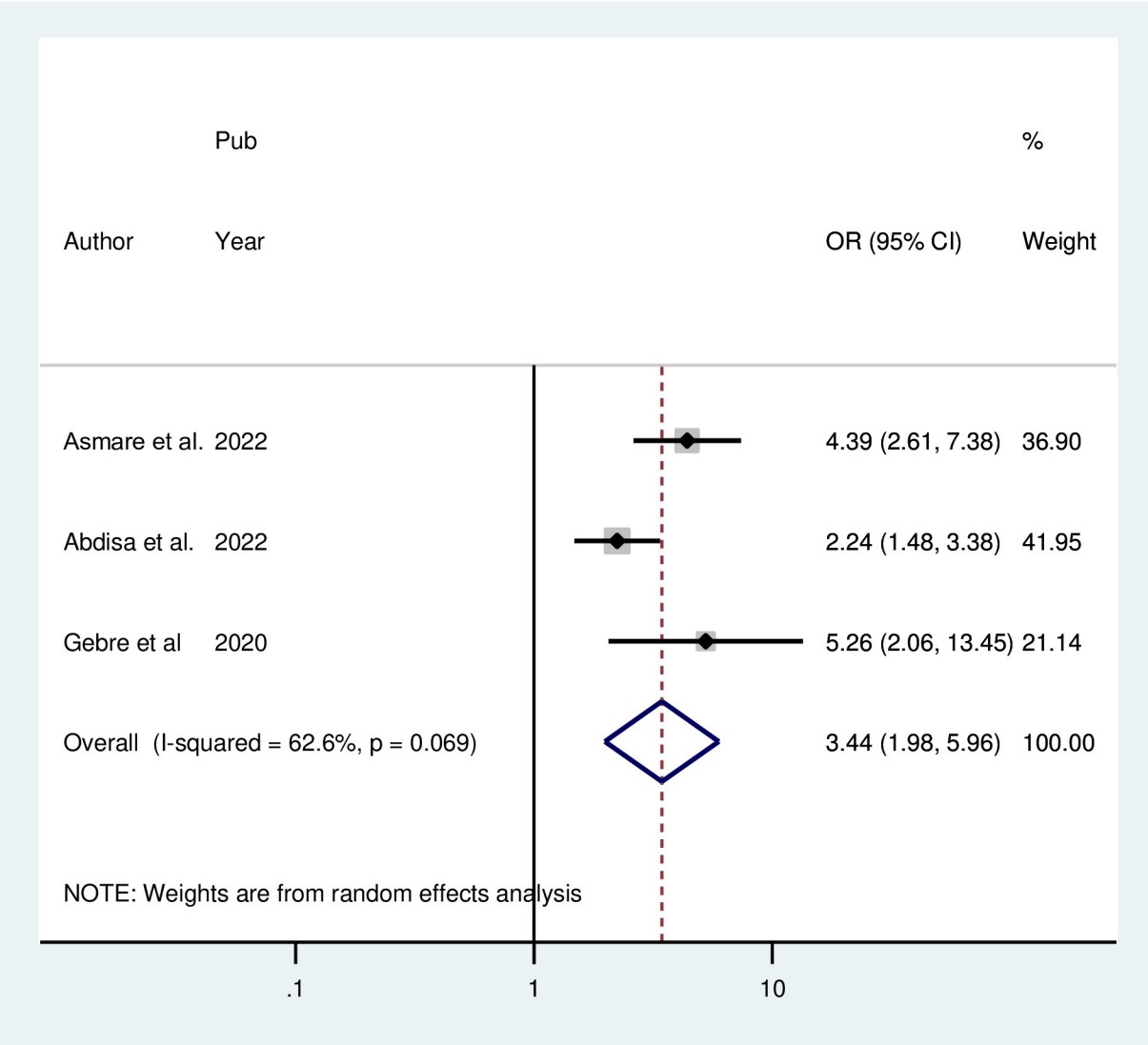

**Fig 8. Forest plot for the association between family history of depression and depression among hypertensive patients in Ethiopia.**

## Conclusion

Overall, the results of our review showed that depression affects a significant number of Ethiopians who have hypertension. For those with hypertension, it makes sense to combine the option of intervention with a mental health assessment. The following factors are significantly associated with depression: being female, being single, having comorbidities, having poor blood pressure control, having a family history of depression, and having poor social support. For those who are depressed, improving the psycho-behavioral treatment linkage with the psychiatric unit can result in improved clinical outcomes. An improved treatment response and overall prognosis for patients with depression depend on early detection and intervention. People are less likely to develop severe or persistent symptoms the earlier they receive the right therapy.

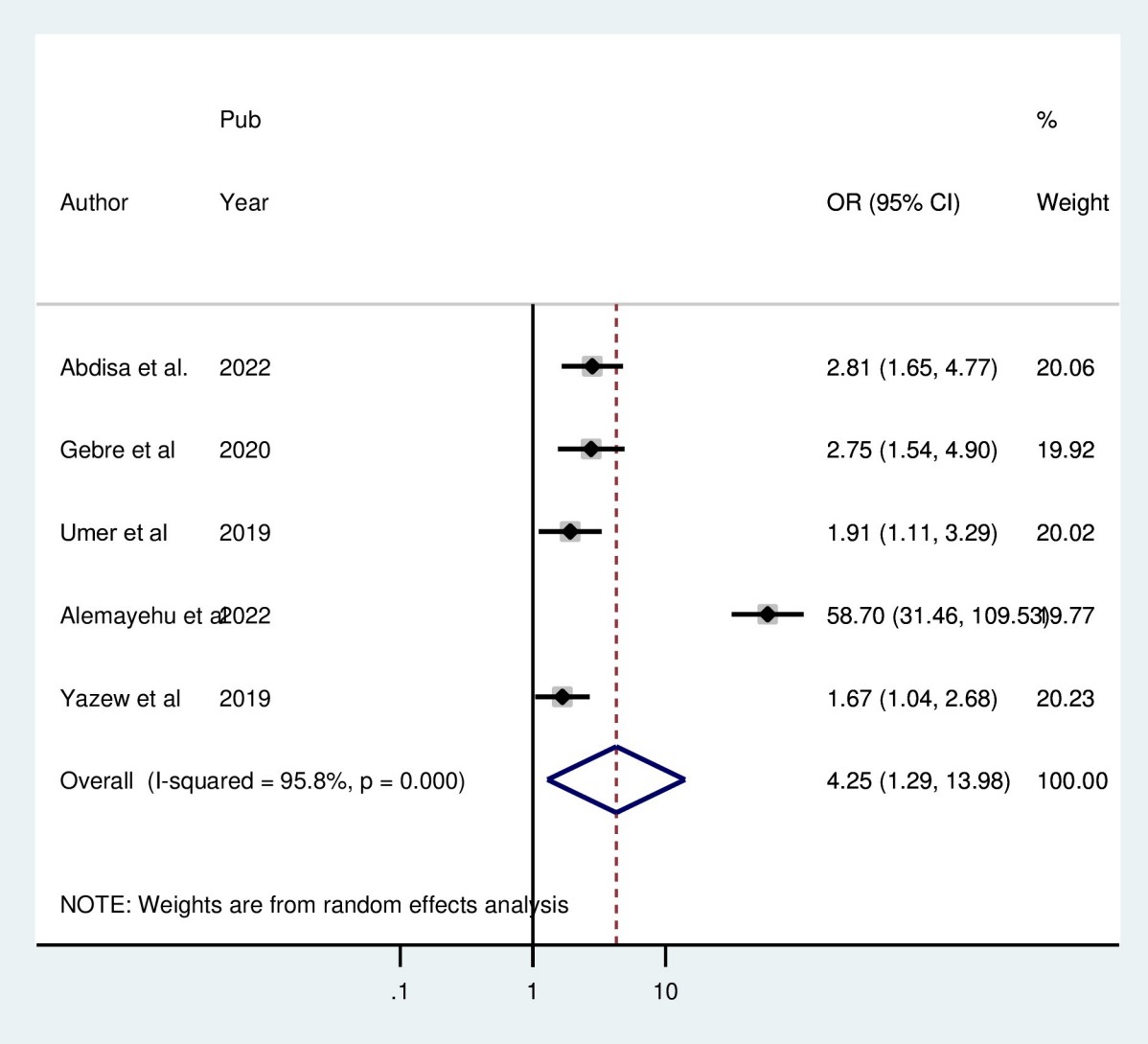

**Fig 9. Forest plot for the association between poor social support and depression among hypertensive patients in Ethiopia.**

## Supporting information

**S1 File. Preferred Reporting Items for Systematic Reviews and Meta-Analyses (PRISMA) checklist.**
(DOC)

**S2 File. Quality assessment of included articles using Newcastle—Ottawa Scale (NOS).**
(DOC)

## Acknowledgments

The authors would like to thank the authors of the included primary studies, which were used as sources of information to conduct this systematic review and meta-analysis.

## Author Contributions

**Conceptualization:** Worku Chekol Tassew.

**Data curation:** Worku Chekol Tassew.

**Formal analysis:** Worku Chekol Tassew.

**Funding acquisition:** Worku Chekol Tassew, Getanew Kegnie Nigate.

**Investigation:** Worku Chekol Tassew, Getaw Wubie Assefa.

**Methodology:** Worku Chekol Tassew, Agerie Mengistie Zeleke.

**Project administration:** Worku Chekol Tassew, Yeshiwas Ayal Ferede.

**Resources:** Worku Chekol Tassew.

**Software:** Worku Chekol Tassew.

**Supervision:** Worku Chekol Tassew.

**Validation:** Worku Chekol Tassew.

**Visualization:** Worku Chekol Tassew.

**Writing – original draft:** Worku Chekol Tassew.

**Writing – review & editing:** Worku Chekol Tassew, Getanew Kegnie Nigate, Getaw Wubie Assefa, Agerie Mengistie Zeleke, Yeshiwas Ayal Ferede.

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
