## [Decision Letter · Decision Letter 0]

24 Apr 2024

PONE-D-24-04651Systematic review and meta-analysis on the prevalence and associated factors of depression among hypertensive patients in EthiopiaPLOS ONE

Dear Dr. tassew,

Thank you for submitting your manuscript to PLOS ONE. After careful consideration, we feel that it has merit but does not fully meet PLOS ONE’s publication criteria as it currently stands. Therefore, we invite you to submit a revised version of the manuscript that addresses the points raised during the review process.

**Please note that we have only been able to secure a single reviewer to assess your manuscript. We are issuing a decision on your manuscript at this point to prevent further delays in the evaluation of your manuscript. Please be aware that the editor who handles your revised manuscript might find it necessary to invite additional reviewers to assess this work once the revised manuscript is submitted. However, we will aim to proceed on the basis of this single review if possible. **

We look forward to receiving your revised manuscript.

Kind regards,

Vanessa Carels

Staff Editor

PLOS ONE

Journal Requirements:

4. Please amend the manuscript submission data (via Edit Submission) to include authors Dr. Getanew Kegnie Nigate, Dr. Getaw Wubie Assefa, Dr. Agerie Mengistie Zeleke and Dr. Yeshiwas Ayal Ferede.

Reviewers' comments:

Reviewer's Responses to Questions

**Comments to the Author**

1. Is the manuscript technically sound, and do the data support the conclusions?

Reviewer #1: Yes

2. Has the statistical analysis been performed appropriately and rigorously? 

Reviewer #1: Yes

3. Have the authors made all data underlying the findings in their manuscript fully available?

Reviewer #1: Yes

4. Is the manuscript presented in an intelligible fashion and written in standard English?

Reviewer #1: Yes

5. Review Comments to the Author

**Reviewer #1: **The manuscript "Systematic review and meta-analysis on the prevalence and associated factors

of depression among hypertensive patients in Ethiopia" investigates the prevalence and factors associated with depression among hypertensive patients in Ethiopia through a systematic review and meta-analysis. The researchers collected data from various databases and analyzed it using statistical methods. The findings showed that the pooled prevalence of depression among hypertensive patients in Ethiopia was 32.43%. Factors associated with depression included being female, having comorbid illnesses, poor blood pressure control, a family history of depression, being single, and poor social support. The study suggests that improving the linkage between psychobehavioral treatment and psychiatric units could lead to better clinical outcomes for depressed hypertensive patients.

Here is my feedback:

1. This systematic review demonstrates exemplary writing and methodological rigor. However, minor adjustments warrant attention.

2. “For those who are depressed, improving the psycho-behavioral treatment linkage with the psychiatric unit can result in improved clinical outcomes” Kindly append a period (.) to the conclusion of the abstract.

3. The authors have presented a thorough literature review. However, it is necessary to incorporate a paragraph discussing the bidirectional impact of depression and hypertension, particularly drawing from the findings of studies conducted in low- and middle-income countries (LMICs).

4. Figure 1 requires revision and redesign as it is of poor quality. Additionally, adjusting the left margin of the figure will enhance its clarity and understanding.

5. In the discussion section, kindly conduct a comparative analysis of your findings with those obtained from other low-income countries (LICs) such as Afghanistan.

6. PLOS authors have the option to publish the peer review history of their article (what does this mean?). If published, this will include your full peer review and any attached files.

Reviewer #1: No

---

## [Author Response · Author response to Decision Letter 0]

29 Apr 2024

April 24 /2024

PONE-D-24-04651

Dear; Editor and Reviewers

Please accept our revised manuscript and note our point-by point response to reviewers below for the manuscript entitled “Systematic review and meta-analysis on the prevalence and associated factors of depression among hypertensive patients in Ethiopia”. Our revised manuscript continues to meet the journal’s formal requirements.

We look forward to your reply and revised decision on the manuscript.

Firstly, we wish to express our great appreciation to the Editorials and the reviewers for your invaluable inputs, commitment, timely manuscript decision and comments, we believe that we have now strengthened our paper based on your comments. Please inspect below a point by point response to the comments raised by reviewers.

N.B Authors’ responses are indicated in bold italics 

Comments from the Editors:

Journal Requirements:

Authors’ response: Thank you for your great comments on the manuscript preparation requirements. We have tried to revise our manuscript and rewrite it again by using PLOS ONE style and improving it.

Authors’ response: Thank you for your great concern. We have corrected our data availability statement to “All relevant data are within the manuscript and its supporting information files,” and we have provided supplementary files for the study

3. PLOS requires an ORCID iD for the corresponding author in Editorial Manager on papers submitted after December 6th, 2016.

Authors’ response: Thank you for your reminder on ORCID ID. I have linked my ORCID ID to Editorial Manager.

4. Please amend the manuscript submission data (via Edit Submission) to include authors Dr. Getanew Kegnie Nigate, Dr. Getaw Wubie Assefa, Dr. Agerie Mengistie Zeleke and Dr. Yeshiwas Ayal Ferede

Authors’ response: Thank you for your consideration of author amendments. We have amended the authors during the submission process.

5. Please review your reference list to ensure that it is complete and correct

Authors’ response: Thank you for your great concern and comments. We have tried to revise our manuscript references, but there are no cited papers that have been retracted currently 

Reviewer's Responses to Questions

Comments to the Author

Reviewer 1: 

1. Is the manuscript technically sound, and do the data support the conclusions?.

Reviewer #1: Yes

Authors’ response: Thank you for your positive comments on the soundness of the manuscript

2. Has the statistical analysis been performed appropriately and rigorously?

Reviewer #1: Yes

 Authors’ response: Thank you for your great feedback on the appropriateness of the statistical analysis

3. Have the authors made all data underlying the findings in their manuscript fully available? 

 Reviewer #1: Yes

Authors’ response: We greatly appreciate the reviewer’s efforts to carefully review the manuscript and provide this answer. 

4. Is the manuscript presented in an intelligible fashion and written in standard English?

 Reviewer #1: Yes

Authors’ response: Thank you very much. This takes into account the improvement of the English language for acceptability and understandability for readers. We tried to correct sentences that made others confused or needed language correction and grammatical errors as well in the revised manuscript

5. Review Comments to the Author

Reviewer #1: The manuscript "Systematic review and meta-analysis on the prevalence and associated factors of depression among hypertensive patients in Ethiopia" investigates the prevalence and factors associated with depression among hypertensive patients in Ethiopia through a systematic review and meta-analysis. The researchers collected data from various databases and analyzed it using statistical methods. The findings showed that the pooled prevalence of depression among hypertensive patients in Ethiopia was 32.43%. Factors associated with depression included being female, having comorbid illnesses, poor blood pressure control, a family history of depression, being single, and poor social support. The study suggests that improving the linkage between psycho-behavioral treatment and psychiatric units could lead to better clinical outcomes for depressed hypertensive patients.

Here is my feedback:

1. This systematic review demonstrates exemplary writing and methodological rigor. However, minor adjustments warrant attention.

Authors’ response: Thank you very much for these great comments. We, as authors, revise and improve our manuscripts based on your comments. Please see the track change manuscript for corrections and modifications. 

2. “For those who are depressed, improving the psycho-behavioral treatment linkage with the psychiatric unit can result in improved clinical outcomes” Kindly append a period (.) to the conclusion of the abstract.

Authors’ response: We wish to express our sincere appreciation to the reviewers for their constructive comments. We have correctly appended the period (.) based on your comments. Thank you again for your constructive and critical comments to improve our manuscript

3. The authors have presented a thorough literature review. However, it is necessary to incorporate a paragraph discussing the bidirectional impact of depression and hypertension, particularly drawing from the findings of studies conducted in low- and middle-income countries (LMICs).

Authors’ response: Thank you very much for your invaluable comments on this part. The discussion is revised again by adding your comments on the bidirectional impact of depression and hypertension and retrieving other study findings

4. Figure 1 requires revision and redesign as it is of poor quality. Additionally, adjusting the left margin of the figure will enhance its clarity and understanding.

Authors’ response: Thank you very much for your invaluable comments to improve figure quality. We have adjusted the margin of the figure and used the Preflight Analysis and Conversion Engine (PACE) digital diagnostic tool to improve the quality and clarity of our manuscript figures.

5. In the discussion section, kindly conduct a comparative analysis of your findings with those obtained from other low-income countries (LICs) such as Afghanistan.

Authors’ response: Thank you very much for your invaluable comments to improve the discussion part. We have performed a comparison or discussed the prevalence of depression in our study with Afghanistan by searching articles. Thank you again for your comments

6. PLOS authors have the option to publish the peer review history of their article (what does this mean?). If published, this will include your full peer review and any attached files.

Authors’ response: Thank you very much for your invaluable comments to improve the discussion part. Sorry for this; we have corrected it based on your comments.

---

## [Editor Report · Decision Letter 1]

6 May 2024

Systematic review and meta-analysis on the prevalence and associated factors of depression among hypertensive patients in Ethiopia

PONE-D-24-04651R1

Dear Dr. tassew,

We’re pleased to inform you that your manuscript has been judged scientifically suitable for publication and will be formally accepted for publication once it meets all outstanding technical requirements.

Kind regards,

Ahmad Neyazi

Guest Editor

PLOS ONE
---

## [Editor Report · Acceptance letter]

13 Jun 2024

PONE-D-24-04651R1 

PLOS ONE

Dear Dr. tassew, 

I'm pleased to inform you that your manuscript has been deemed suitable for publication in PLOS ONE. Congratulations! Your manuscript is now being handed over to our production team.

Kind regards, 

on behalf of

Dr. Ahmad Neyazi 

Guest Editor

PLOS ONE